# Structure of *Chlorella ohadii* Photosystem II Reveals Protective Mechanisms against Environmental Stress

**DOI:** 10.3390/cells12151971

**Published:** 2023-07-31

**Authors:** Maria Fadeeva, Daniel Klaiman, Ido Caspy, Nathan Nelson

**Affiliations:** Department of Biochemistry and Molecular Biology, The George S. Wise Faculty of Life Sciences, Tel Aviv University, Tel Aviv 69978, Israel; mariiaf@tauex.tau.ac.il (M.F.); klaiman79@gmail.com (D.K.); idocaspy@mail.tau.ac.il (I.C.)

**Keywords:** green algae, cryoEM, photosynthesis, photosystem II, photoprotection, oxygen evolving complex, structure

## Abstract

Green alga *Chlorella ohadii* is known for its ability to carry out photosynthesis under harsh conditions. Using cryogenic electron microscopy (cryoEM), we obtained a high-resolution structure of PSII at 2.72 Å. This structure revealed 64 subunits, which encompassed 386 chlorophylls, 86 carotenoids, four plastoquinones, and several structural lipids. At the luminal side of PSII, a unique subunit arrangement was observed to protect the oxygen-evolving complex. This arrangement involved PsbO (OEE1), PsbP (OEE2), PsbB, and PsbU (a homolog of plant OEE3). PsbU interacted with PsbO, PsbC, and PsbP, thereby stabilizing the shield of the oxygen-evolving complex. Significant changes were also observed at the stromal electron acceptor side. PsbY, identified as a transmembrane helix, was situated alongside PsbF and PsbE, which enclosed cytochrome *b559*. Supported by the adjacent C-terminal helix of Psb10, these four transmembrane helices formed a bundle that shielded cytochrome *b559* from the surrounding solvent. Moreover, the bulk of Psb10 formed a protective cap, which safeguarded the quinone site and likely contributed to the stacking of PSII complexes. Based on our findings, we propose a protective mechanism that prevents Q_B_ (plastoquinone B) from becoming fully reduced. This mechanism offers insights into the regulation of electron transfer within PSII.

## 1. Introduction

Oxygenic photosynthesis of cyanobacteria, various algae, and land plants converts light energy from the sun into a biologically useful chemical energy concomitant with the evolution of molecular oxygen [1,2]. The light reactions of photosynthesis depend on the functions of photosystem II (PSII), cytochrome b6f, and photosystem I (PSI). Driven by photon energy, these membrane-embedded machineries carry out a linear electron transfer from water to NADP^+^, and eventually, reduce CO_2_ to form organic material [3]. PSI exhibits a different structural organisation among the photosynthetic organisms, while PSII organises itself mainly into a dimer, regardless of the species of the organism. However, PSII interacts with a variety of light-harvesting proteins, to form supercomplexes of different sizes and shapes [4]. In *Chlamydomonas reinhardtii*, the various supercomplexes include C2S2 and C2S2M2L2 (C indicates the PSII core, while S, M, and L indicate strongly, moderately, and loosely associated LHCII, respectively). Among these, C2S2M2L2 is the largest known PSII–LHCII supercomplex in green algae or plants [4,5]. So far, none of the published structures of PSII contain all the expected subunits of the complex.

Performing photosynthesis under extreme high light (HL) intensities is a challenge for photosynthetic organisms, as well as for the artificial systems that rely on natural photosystems. As such, specialised photoautotrophs have devised mechanisms to dissipate the excess excitation energy and reduce the potential photodamage that intense illumination causes to PSII, and also to the less sensitive Photosystem I (PSI) [6,7]. The green alga *Chlorella ohadii*, which was isolated from a desert biological soil crust, copes with these harsh conditions, including extremely high daytime illumination (~2000 μE) [8]. Unlike other photosynthetic organisms, *C. ohadii* does not undergo photodamage, even when exposed to light intensities that are three to four-fold higher than that which is required to saturate the CO_2_ fixation [9]. It has been proposed that under high illumination, *C. ohadii* activates several protection mechanisms, including a massive cyclic electron transport within PSII that can be as high as 90% of the electrons from water splitting [10,11]. These adaptive capabilities should be reflected in structural alterations in the PSII complex. We isolated a highly active PSII supercomplex from *C. ohadii* and determined its high-resolution structure (Figure 1, Appendix A) using cryoEM (Methods).

## 2. Materials and Methods

### 2.1. Purification of the PSII Supercomplex from C. ohadii Cells

The *C. ohadii* cells were cultured in 10 L of TAP (Tris-Acetate-Phosphate) minimal medium under continuous white light (80 µE) at 25 °C for about 4 days, until they reached an absorbance of 0.75 OD at 730 nm. The culture was harvested by centrifugation at 3500× *g* for 5 min and re-suspended in a medium containing 25 mM MES-KOH, pH = 6.0, 200 mM sucrose, 10 mM NaCl, and 5 mM MgCl_2_. The cells were washed once in the same buffer, spun down by centrifugation at 5000× *g* for 5 min, and then suspended in a homogenization buffer containing 30 mM MES–NaOH, pH = 6.0, 300 mM sucrose, and 0.2 mM EDTA-Na. Protease-inhibitor cocktail was added to the final concentrations of 1 mM PMSF, 1 µM pepstatin, 60 µM bestatin, and 1 mM benzamidine. The cells were disrupted by an Avestin^®^ EmulsiFlex-C3 at 2000 psi (two cycles), and the resulting suspension was immediately diluted twice with a buffer containing 20 mM MES–NaOH, pH = 6.0, and 300 mM sucrose, to decrease the EDTA concentration. The unbroken cells and starch granules were removed by centrifugation at 12,000× *g* for 5 min, and the membranes in the supernatant were precipitated by centrifugation in a Ti70 rotor at 148,200× *g* for 40 min. The pellet was suspended in the buffer that contained 20 mM MES–NaOH, pH = 6.0, and 300 mM sucrose, giving a chlorophyll concentration of 2.0 mg/mL.

n-Decyl-α-D-Maltopyranoside (α-DM) and n-octyl β-D-glucopyranoside were added dropwise to a final concentration of 2.0% and 0.75%, respectively. After stirring at 4 °C for 25 min, the insoluble material was removed by centrifugation at 20,800× *g* for 15 min. The supernatant was loaded onto the sucrose gradients in an SW-60 rotor (≈640 µg of chlorophyll per tube). The gradient composition was 20–50% sucrose, 20 mM MES–NaOH, pH 6.0, and 0.2% αDM, and this was centrifuged at 310,000× *g* for 14–16 h. Appendix A shows the distribution of the green bands in the tubes and the SDS-PAGE of the main bands. The band containing PSII was diluted ten-fold to reduce the sucrose, using 20 mM MES–NaOH, 6.0, and 0.1% α-DM to remove the sucrose, and concentrated using the centrifugal concentrator Vivaspin^®^6 (MWCO 100,000 PES membrane), to a final chlorophyll concentration of 2.9 mg/mL, with an oxygen evolution activity of 336 µmole O_2_/mg chl/h.

### 2.2. CryoEM and Image Processing

Three µL of purified PSII at a concentration of 3 mg/mL was applied to the glow-discharged holey carbon grids (Cu QUANTIFOIL^®^ R1.2/1.3), then vitrified using a Leica GP2 plunge freezer (3 s blot, 20 °C, 100% humidity). The images were collected on a 300 kV FEI Titan Krios2 electron microscope (EMBL, Heidelberg, Germany). A Gatan K3 Summit detector was used in counting mode at a magnification of 105,000 (yielding a pixel size of 0.64 Å), with a total dose of 50.6 e/Å^2^. Thermo Fisher Scientific EPU software (Titan Krios G3) was used to collect a total of 21,341 images, which were dose fractionated into 40 movie frames, with defocus values ranging from 0.8 to 1.8 μm, at 0.1 μm increments. The collected frames were motion-corrected and dose-weighted using MotionCor2 [12]. The contrast transfer function parameters were estimated using Ctffind-4.1 [13]. A total of 734,768 particles were picked using reference-free picking in RELION-v.3.1 [14]. The picked particles were processed for reference-free 2D averaging, resulting in 508,108 particles for the initial model building; both steps were performed when using RELION. Following the initial model creation, the 2D subset was used for the 3D classification, resulting in three distinct classes in RELION. From these, the best class was selected, which contained a total of 379,578 particles. These particles were pooled together and processed for 3D homogeneous refinement and postprocessing using RELION. The reported 2.72 Å resolution of PSII was based on a gold-standard refinement, applying the 0.143 criteria on the FSC between the reconstructed half-maps (Appendix A).

## 3. Results and Discussion

We isolated a highly active PSII supercomplex from *C. ohadii* and determined its high-resolution structure (Figure 1, Appendix A) using cryoEM (Methods). Overall, the structure resembled that of the C2S2M2L(N)2-type PSII-LHCII supercomplex from *C. reinhardtii* (PDBID 6KAD and 6KAF) [4,15]. Our new structure contains three additional subunits in each PSII monomer, denoted as PsbP, Psb10, and PsbY (Figure 1, Figure 2 and Appendix A), which might serve as pillars for the stability of the *C. ohadii* complex (PDBID 8BD3). Moreover, subunits that are critical for the stability and the oxygen evolution of PSII are better resolved, exhibiting critical interaction sites with other subunits of the complex. Appendix A illustrates the differences between the structures of *C. ohadii* (PDB 8BD3) and the equivalent *C. reinhardtii* structure (PDBID 6KAD).

The cryoEM structure of PSII at 2.72 Å unveiled 64 subunits containing 386 chlorophylls, 86 carotenoids, four plastoquinones, and several conserved structural lipids. The PSII reaction centre is engulfed by shielding subunits, both from the luminal and stromal side (Figure 2 and Figure 3). The function of these extrinsic luminal subunits of PSII is still under strong debate [16,17,18]. In higher plants, they have been proposed to operate primarily during the biogenesis of the complex [17]. The main conclusion from the numerous reported experiments is that these domains are essential for the stability of the PSII supercomplex. The structure of the luminal extrinsic subunits and their interactions have been elucidated (Figure 2). It is well documented that Oxygen Enhancer Element 1 PsbO (OEE1) intactness is critical for the stability and the activity of PSII. A single amino acid substitution (P104H) in *C. reinhardtii* has rendered PSII temperature sensitive and caused the disappearance of the entire complex at the nonpermissive temperature of 37 °C [19]. The tertiary structure of *C. ohadii* PsbO is quite conserved, sharing 75% identity with the sequence of the *C. reinhardtii* protein. *C. ohadii* PsbO is secured by strong interactions with CP47 via a stabilizing loop that was not resolved in the *C. reinhardtii* structure, together with subunit PsbP, that is not present in the latter structures (PDBID 6KAD and 6KAF) [4,15].

PsbP is an Oxygen Enhancer Element 2 (OEE2) that was pronouncedly present in our structure (Figure 1 and Figure 2). PsbP deletion in *C. reinhardtii* resulted in the loss of oxygen evolution activity but was only marginally affected in the assembly of the other PSII subunits [20]. Regardless of its expression in the various organisms, PsbP was missing in several of the PSII preparations. In *C. reinhardtii*, PsbP was identified in the cryoEM structure of the C2S2-type PSII-LHCII supercomplex from (PDB 6KAC) [4], however, it was not present in the structure of the larger PSII form (PDB 6KAD) [15]. The *C. ohadii* PsbP counterpart was only 69% identical to its *C. reinhardtii* homologue. Yet, despite the differences in the amino acid sequences, the structure of the corresponding subunits was relatively similar. Thus, the interaction between the specific amino acids was probably responsible for the stability that *C. ohadii* OEE2 conferred. The tight binding of PsbP might contribute to the overall stability of the large luminal structure that protects OEC.

The sequence of the nuclear gene of the 16 kDa Oxygen Enhancer Element 3 (OEE3) polypeptides present in the oxygen evolving complex of *C. reinhardtii* have been established [21,22]. The comparison between the OEE3 protein sequences of *C. reinhardtii* and the higher plants has revealed a mere 28% overall homology, mostly limited to the central portion of the protein. We identified OEE3 in the *C. ohadii* PSII structure (PsbU) as four bundled helices laying atop CP43, contacting PsbO (Figure 2). PsbU extended N-terminus contacted with PsbP, supported by a subsequent helix–helix interaction with CP43 (Figure 2). This arrangement has suggested a major function in stabilising the entire large complex, isolating the OEC from the luminal medium. The PsbU subunit, like PsbP, was absent in the large PSII complex from *C. reinhardtii* (PDBID 6KAD).

Due to the formation of stacked PSII in the grana, there was no place for large protrusions at the stromal side of the complex. So far, no specific amino acid sequences were shown to be directly involved in the PSII dependent grana formation [23]. The structure of *C. ohadii* PSII presented a slight deviation to this rule by the presence of the “PSII 10 kDa polypeptide”. This subunit has been denoted as Psb10 and was evident in our structure (Figure 2 and Figure 3). An unidentified stromal protein (USP) that was present in the C2S2 supercomplex but was not observed in the C2S2M2L2 supercomplex of *C. reinhardtii* might be its homologue [4]. Subunit Psb10 was located at the interface between two adjacent C2S2 supercomplexes that are stacked with each other along their stromal surface and might represent the partial Psb10 subunit that is entirely present in the *C. ohadii* structure (Figure 3). According to the published sequences, Psb10, besides algae, was present also in mosses and higher plants—but not in cyanobacteria. This suggests that Psb10 might have emerged alongside the PSII and mediated grana stacking. The Psb10 polypeptide N-terminus was located at the stroma, forming a cap over the interface between CP43 and D2, subsequently establishing an interface with PsbE and the PsaF N-termini. Finally, it formed a transmembrane helix parallel to and in strong contact with the PsbE transmembrane helix (Figure 3). The position of this helix was similar to that of the alpha helix that was identified in PDB 6KAC and built as a polyalanine chain [4]. The Psb10 C-terminus interacted with PsbP on the luminal side of the membrane (Figure 2 and Figure 3). This arrangement leaves little doubt regarding the important function of Psb10 in stabilising the *C. ohadii* PSII supercomplex. Finally, Psb10 might support the interaction between the two PSII complexes that form the stacked PSII (Appendix A). The function of Psb10 is unknown but its four tyrosine residues facing the Q_A_-Fe-Q_B_ cluster suggest an involvement in protection against radicals (Figure 2, Figure 3 and Figure 4).

Another pillar connected to the subunits PsbE, PsbF, and Psb10 was in the position of PsbY in the PSII complex. PsbY is a single transmembrane helix protein, yet it is encoded and translated as multiple copies in a single protein. In *C. ohadii*, PsbY appeared in four copies, but in some green algal species, such as *Micractinium conductrix*, *Volvox*, *Chlamydomonas*, and *Scenedesmus*, it manifested itself in three to five copies, while in the higher plants, only one or two copies were present in a single transcript. The copies within the transcript were identical, or highly homologous. Polyprotein transcripts are quite common in viruses [24], and it would be of no surprise if PsbY is of viral origin. This notion was supported by the different copies present in the closely related organisms (Appendix A). The variation in the number of PsbY copies between different plant species suggests a potential difference in the functional requirements and regulatory mechanisms of PSII among plant lineages. The presence of multiple PsbY subunits in algae and lower plants may reflect specific adaptations and variations in the assembly or stabilization of the PSII complex in these organisms.

PsbY joined a triplet of transmembrane helices that were formed by PsbE, PsbF, and Psb10, while running antiparallel to them (Figure 3c). Together, they shield the heme group of cytochrome *b559*, which otherwise would be exposed to the membrane interior environment and could easily contact with potential deleterious electron acceptors. Thus, PsbY and Psb10 provide additional protection to that of PsbE and PsbF. This arrangement stabilized the position of the prosthetic group that might be crucial for the protection of PSII from photodamage caused by excessive light intensities, when electron acceptors are lacking. So, even increased CEF during high light illumination does not lead to increased production of free radicals and excess damage of PSII with subsequent inactivation.

Following light absorption, charge separation converts excited P680* to create the longer-lived state Yz^+^ Q_A_^−^. The hole in Yz^+^ can be filled by an electron from the manganese oxygen evolving complex (OEC), or from the electron acceptor side, either directly, or via rapid equilibrium with P680^+^. Under ambient conditions, Q_A_^−^ transfers its electron forward to Q_B_, which is mediated by the non-heme iron site [25]. The reaction is relatively slow and takes place on a sub millisecond time range [26]. If the oxidized quinones are absent, or present in limited availability, the electron may return via many possible routes to P680^+^ or OEC. None of these routes are visible under continuous light, or even via the S-state turnover measurements [27,28,29] and it was considered to be radical protective cyclic electron transport [28,29,30]. Cytochrome *b559* was suggested to participate in this cyclic electron transport [30]. The presence of oxidized PQ in tiny amounts, which was maintained by the cycling electrons, might be sufficient to protect this photosynthetic organism from radical damage.

The efficient drainage of electrons/reductants from the reaction centres for metabolic or other uses could help protect the photosynthetic machinery from the damage that is caused by the excess illumination [10]. Q_A_^•−^, the reduced semiquinone form of the nonexchangeable quinone, is often considered capable of a side reaction with O_2_, forming superoxide and damaging radicals [30,31]. Fantuzzi et al., (2022) [32], using chlorophyll fluorescence in plant PSII membranes, showed that O_2_ oxidizes Q_A_^•−^ at physiological O_2_ concentrations, with a t_1/2_ of 10 s. Q_A_^•−^ could only reduce O_2_ when bicarbonate was absent from its binding site on the nonheme iron (Fe^2+^). It was concluded that the reduction of O_2_ was favourable when the oxygen binds directly to Fe^2+^. This contrasts with a previously proposed mechanism involving direct oxidation of Q_A_^•−^ by O_2_, which was expected to require close contact between the oxygen and the semiquinone [32,33,34]. Thus, *C. ohadii* PSII, which copes with extremely high daytime illumination of up to 2000 μE, either contains an extremely tight bicarbonate binding site, or has an altered sequence of events that are linked to the reduction state of the quinone pool [35]. Our structure suggests a conserved bicarbonate binding site. Additionally, in contrast to most preparations of eukaryotic PSII, where Q_B_ is missing, in *C. ohadii* Q_B_, it was clearly identified at its binding site. So far, Q_B_ was clearly seen mostly in significantly more stable PSII preparations from cyanobacteria.

This structure revealed strong interactions with the surrounding amino acids and prosthetic groups (Figure 4a). It is worth noting that the terminal end of its isoprenoid chain was tightly held by the surrounding hydrophobic amino acids. The enhanced binding of Q_B_ and plastoquinones in the hydrophobic pocket of *Chlorella ohadii* PSII leads to several advantageous effects. Firstly, it facilitates the clear detection and localization of Q_B_ within the PSII structure. This knowledge provides valuable insights into Q_B_’s precise role in the electron transfer process during photosynthesis. Secondly, the strong binding of Q_B_ and plastoquinones in *Chlorella ohadii* PSII contributes to its resistance against photodamage induced by high light intensities. Accordingly, the direct electron transfer from Q_A_^−^ to Q_B_ and the fast transfer of the electron from Q_B_^−^ to the quinone pool protects *C. ohadii* PSII from photodamage by removing the bottleneck of electron transfer on the “Q_A_^−^ to Q_B_” stage. In the case of the previously described mechanism, when reduced Q_B_ is proposed to leave and be replaced by another oxidized Q_B_, this step would limit the electron transfer in the PSII part of all the electron-transport chain, and newly arrived electrons will be stuck on the Q_A_^−^ stage, increasing the probability of forming radicals and damaging PSII subunits. The observation that the plastoquinone pool of *Nannochloropsis oceanica* was not completely reduced during the bright light pulses [36], as well as in the plants [37], is in line with this study’s proposal for the mechanism of photoprotection in *C. ohadii*.

A wide hydrophobic cavity was observed, starting at Q_B_ and going out of PSII to the middle of the membrane, and might contain several quinones (Figure 5).

A similar cavity was detected in the crystal structure of *T. elongatus*, and it was proposed to harbour a specialised quinone C [37,38,39,40]. The potential occupancy allowed the quinones to pack together, maintaining distances of less than 10 Å from each other (Figure 4c). This distance allows electron transfer in about 1 ns, which is several orders of magnitude faster than 0.5 ms required for the electron transfer from Q_A_ to Q_B_ [26,41]. Three plastoquinone exchange pathways were previously identified by molecular dynamic simulations of PSII [42]. One of these ensued close to the heme of cytochrome *b559* and might coincide with the hydrophobic cavity that is described in this work. This part of the PSII hydrophobic cavity was exposed to the membrane, providing an ideal binding site (if one even exists) to the cytochrome b6f complex. Our suggested mechanism does not require an instantaneous escape of reduced Q_B_, and the exchange of the reduced and oxidised plastoquinones might take place at the edge of the hydrophobic cavity, or even outside in the mid-membrane medium (Figure 5). Of course, this is not the only possible mechanism of PSII protection plants developed during evolution. Interesting examples of different mechanisms to adapt photosynthetic chain functioning to high-light conditions on the PSII stage is associated with the multifunctional subunit PsbS [43].

*C. ohadii* was far less sensitive to diuron (DCMU) than other green algae [9]. This effect can be explained by the tighter binding of Q_B_ and reduced diuron accessibility to its binding site. A recent study on the binding properties of the photosynthetic herbicides with the Q_B_ site of the D1 protein in plants have demonstrated that the high affinity inhibitors, such as diuron, have replaced Q_B_ at its binding site [41,42,44]. A similar mode of interaction was established for terbutryn binding in cyanobacterial PSII [45,46]. The less potent herbicides, such as bentazon, bind away from this site in a manner that should prevent the secondary quinones from accessing the cavity, as proposed in this manuscript. Thus, a high metabolic rate, such as rapid lipid production, might serve as the best route to protect PSII from the damage that is caused by the intense illumination [37,47]. The accumulation of quinones in the vicinity of the reaction centre (CLA407/A), with Q_B_ and the heme of PsbF, might also provide an ideal environment for cycling the excess electrons, or to quench the radicals when they occur (Figure 4c). This is in line with the suggestion that PsbL prevents the reduction of PSII by the back-electron flow from plastoquinol, protecting PSII from photoinactivation, whereas PsbJ regulates the forward electron flow from Q_A_^*^- to the plastoquinone pool [2].

## 4. Conclusions

In this paper we reported the most complete description of a eukaryotic PSII cryoEM structure from the high-light resistant green algae *C. ohadii*, revealing a tight encapsulation and robust protection of the OEC, providing a structural basis for the algae’s ability to resist photodamage, even at extreme illumination intensities. On the electron acceptor side, we identified a novel subunit Psb10 that together with PsbE, PsbF, and PsbY, shielded cytochrome *b559* from its local environment and also served to increase protection from photodamage. Finally, we detected the elusive Q_B_ in our structure, probably due to the unique architecture of its hydrophobic binding pocket in *C. ohadii* PSII, that leaves ample space for additional quinones to occupy, thus generating a protective mechanism that prevents over-reduction of Q_B_ molecules. The quinone exit cavity of PSII plays a critical role in determining the overall global entropy in living organisms. Building upon our newly determined PSII structure and the demonstration of Q_B_ binding, we have formulated a hypothesis that Q_B_ is infrequently fully reduced and that the electrons rapidly transfer to the oxidized quinones present in the exit cavity. If this hypothesis is substantiated, it would have profound implications for our understanding of the mechanism of PSII. Furthermore, the implications of this finding extend beyond fundamental research. Understanding the precise mechanism of electron transfer in PSII could lead to innovative strategies to improve crop productivity.

## Figures and Tables

**Figure 1 cells-12-01971-f001:**
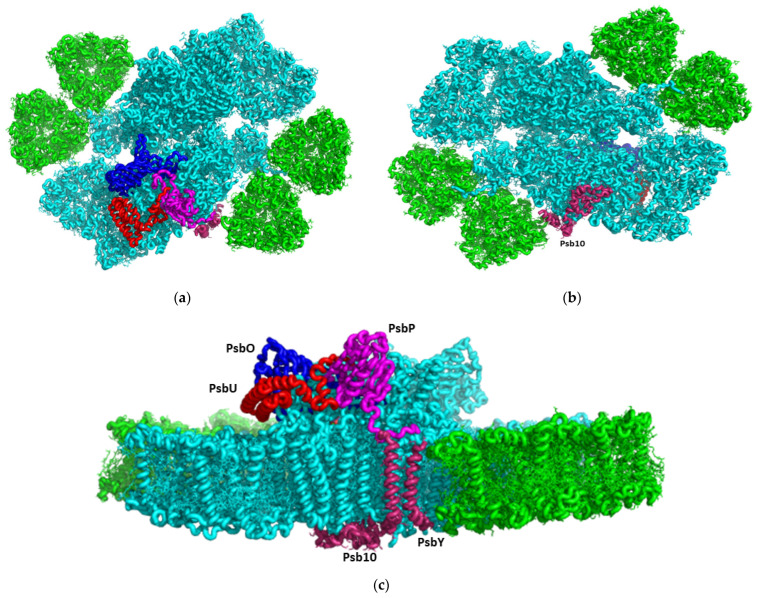
CryoEM structure of the *C. ohadii* PSII complex: (**a**) Top view of the supercomplex from the stromal side. The novel or rare subunits are coloured (PsbP-magenta, PsbO-blue, PsbU-red, and Psb10 and PsbY are in purple; (**b**) Luminal view of the supercomplex; (**c**) Side view along the membrane plane.

**Figure 2 cells-12-01971-f002:**
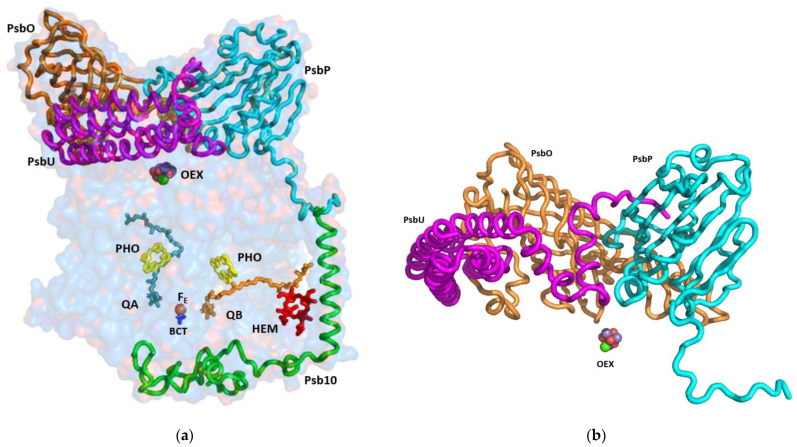
Side view of the subunits that are shielding the reaction centre: (**a**) The indicated subunits and the prosthetic groups are shown on the background of a surface model, with 80% transparency of the PsbA, PsbB, PsbC, PsbD, PsbP, PsbO, PsbU, and Psb10 subunits; (**b**) The structure of the luminal extrinsic subunits and their interactions with each other: A ribbon presentation of the three main subunits protecting the oxygen evolving complex. The extensive interactions among OEE1 (PsbO), OEE2 (PsbP), and OEE3 (PsbU) are shown. OEE—oxygen evolving enhancer; OEX—oxygen evolving complex; PHO—pheophytin, BCT—bicarbonate anion.

**Figure 3 cells-12-01971-f003:**
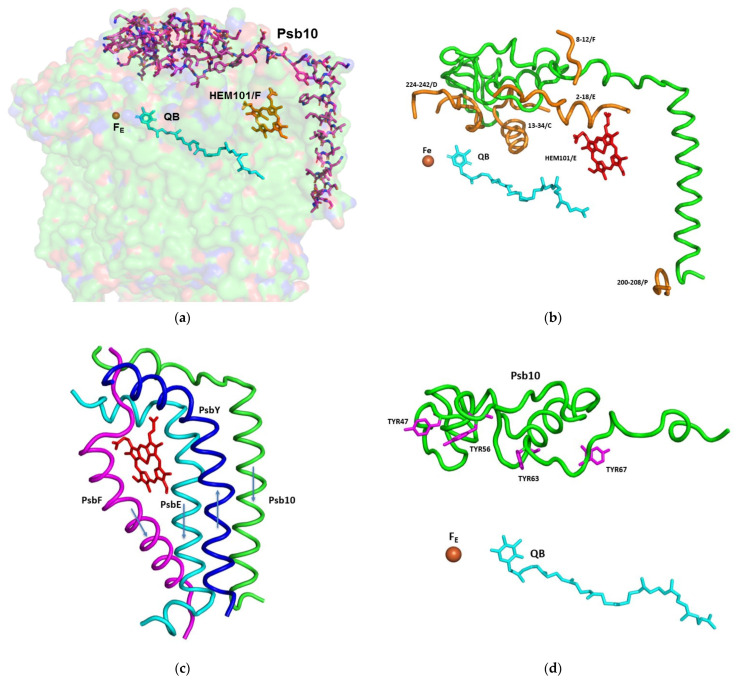
The structure and interaction of the subunits involved in the protection of the acceptor site of PSII, including Fe, Q_B_, and cytochrome *b559*: (**a**) Stick presentation of Psb10 shown on the background of a surface model, with 80% transparency of the subunits PsbA, PsbC, PsbD, PsbE, PsbF, PsbY, and Psb10; (**b**) Interaction of Psb10 (in green) with the various PSII subunits: A ribbon presentation of the Psb10 interaction with the indicated subunits; (**c**) Transmembrane helices shielding the heme of cytochrome *b559*. The transmembrane helices of Psb10, PsbE, and PsbF are oriented from the stroma to the lumen. The helix of PsbY is going in the opposite direction; (**d**) Four tyrosine residues of Psb10 pointing to the Fe-Q_B_ location.

**Figure 4 cells-12-01971-f004:**
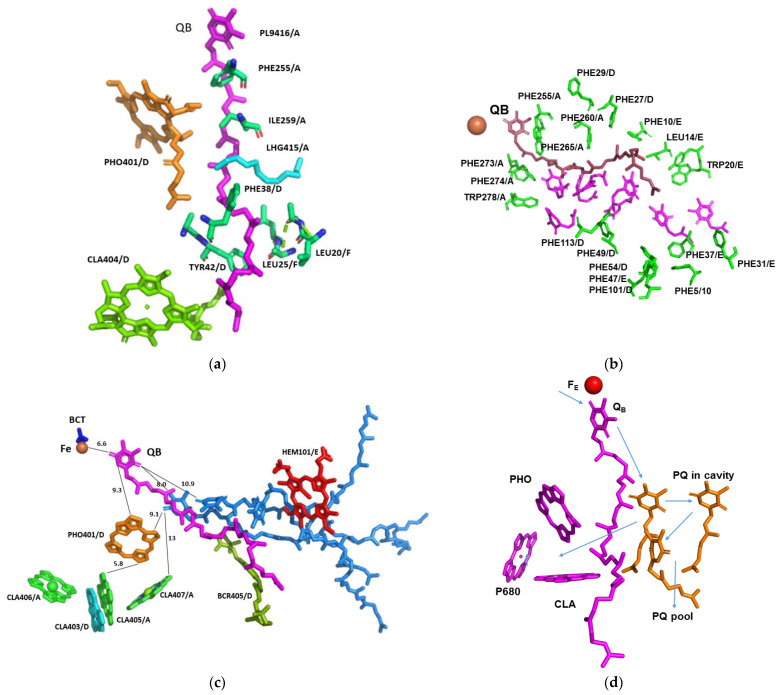
Structure of Q_B_ and its vicinity in relation to the factors involved in the electron transport and quenching: (**a**) Interactions of Q_B_ with the surrounding amino acids and prosthetic groups. The amino acids and the prosthetic groups that are in contact with Q_B_ are specified; (**b**) Hydrophobic cavity proceeding from Q_B_ to the exterior of the complex. The amino acids surrounding the cavity are specified and coloured green. Space filling of seven Q_1_ molecules (magenta) modelled into the cavity with no structural clashes; (**c**) Position and distances of the prosthetic groups pertinent for the function of PSII. Three plastoquinone molecules (light blue) modelled into the cavity with no structural clashes. The indicated distances are in Å; (**d**) Potential electron transfer pathways that might be involved in the electron transfer from Q_B_ to plastocyanin, quenching, and the cyclic electron transport. PHO—pheophytin, BCT—bicarbonate anion; PL9—plastoquinone, LHG—1,2-dipalmitoyl-phosphatidyl-glycerol, BCR—beta-carotene.

**Figure 5 cells-12-01971-f005:**
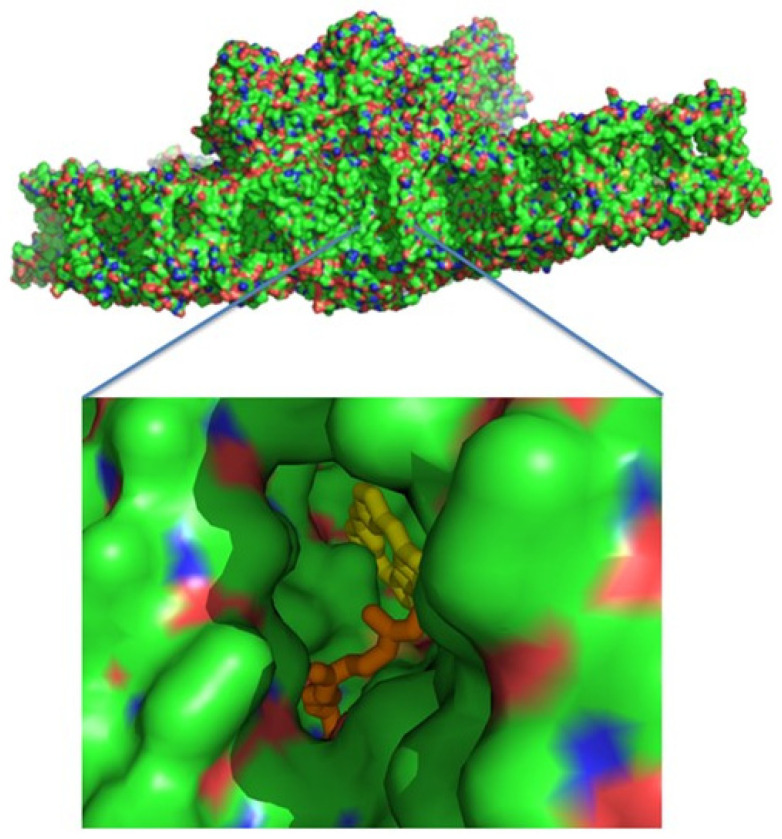
A surface presentation of PSII viewed from the membrane interior. Zooming into the middle reveals a hydrophobic cavity, where pheophytin A and part of the Q_B_ isoprenyl chain are apparent.

## Data Availability

The atomic coordinates of the three supercomplexes have been deposited in the Protein Data Bank, with accession code PDB 8BD3. The cryoEM maps have been deposited in the Electron Microscopy Data Bank, with accession codes PDB entry ID 8BD3 and EMDB entry ID EMD 15973.

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
