# Peer review of "Structure of Chlorella ohadii Photosystem II Reveals Protective Mechanisms against Environmental Stress"

_cells, 2023, doi:10.3390/cells12151971_

Round 1

Reviewer 1 Report

This manuscript is interesting and I would like to recommend it to be accepted by the current journal. However, the manuscript can be better connected with the current journal. 

Author Response

Thank you for the positive review. 

Reviewer 2 Report

This manuscript reports the structure of the PSII complex of  Chlorella ohadii at 2.7 A resolution by cryo-EM. Although the structure of PSII has been reported earlier, this work is highly interesting and important because C. ohadii grows in the desert under very harsh environmental conditions such as extremely high illumination implying that this alga has evolved particular protection mechanisms  against photo-oxidative damage which are still largely unknown. In this respect the structure of PSII  determined in this study provides several new insights. First it reveals three new subunits PsbP, Psb10  and PsbY  in the PSII cryo-EM structure. Second , the structure reveals that the shield of the oxygen-evolving complex on the lumen side is stabilized through a unique arrangement of the PsbO, PsbP, PsbB and PsbU (OEE3) with PsbU interacting with PsbO, PsbC and PsbP. Third, at the PSII stromal electron acceptor side, cytochrome b559 is shielded by a  bundle of the four transmembrane helices  of psbY, PsbF, PsbE and Psb10 with Psb10 forming a protective cap around the elusive quinone B site, first detected here, which is proposed to prevent QB over-reduction.

One aspect of the manuscript which should be improved is to provide a clearer description of the difference between the PSII structure of C. reinhardii and  C. ohadii. In this respect the manuscript is difficult to  follow. The authors could show a supplementary Fig showing side by side the structures of the two algae and pinpointing the differences.

Additional remarks

l.92 replace PSI by PSII

L.161 replace  In C. reinhardtii, PSII … by In C. reinhardtii, PsbP….

L.282 reveilles? reveals?

L.162,163 “One example is a R368F substitution 16Å away of 167 the oxygen evolving complex (OEC). A nearby substitution of K371G might also contribute to the stability of this important site.” These substitutions refer presumably to C. reinhardtii vs C. ohadii. This should be stated more clearly in the text

Fig. 3 Coiors of the subunits and cofactors change from one panel to the other even in the same Figure. It would be less confusing for the reader to use the same color code for all Figures.

Some abbreviations in the  Figures need to be explained in the legends, e.g. PHO, BCT, PL, LHG

Minor corrections of some sentences in the text are needed.

Author Response

Comments and Answers

Comment

This manuscript reports the structure of the PSII complex of Chlorella ohadii at 2.7 A resolution by cryo-EM. Although the structure of PSII has been reported earlier, this work is highly interesting and important because C. ohadii grows in the desert under very harsh environmental conditions such as extremely high illumination implying that this alga has evolved particular protection mechanisms against photo-oxidative damage which are still largely unknown. In this respect the structure of PSII determined in this study provides several new insights. First it reveals three new subunits PsbP, Psb10 and PsbY in the PSII cryo-EM structure. Second, the structure reveals that the shield of the oxygen-evolving complex on the lumen side is stabilized through a unique arrangement of the PsbO, PsbP, PsbB and PsbU (OEE3) with PsbU interacting with PsbO, PsbC and PsbP. Third, at the PSII stromal electron acceptor side, cytochrome b559 is shielded by a bundle of the four transmembrane helices of psbY, PsbF, PsbE and Psb10 with Psb10 forming a protective cap around the elusive quinone B site, first detected here, which is proposed to prevent QB over-reduction. One aspect of the manuscript which should be improved is to provide a clearer description of the difference between the PSII structure of C. reinhardii and C. ohadii. In this respect the manuscript is difficult to follow. The authors could show a supplementary Fig showing side by side the structures of the two algae and pinpointing the differences.

Answer:

We express our gratitude to the reviewer for their prompt and meticulous revision of our paper. We acknowledge that our intention to present the ideas in a concise form may have made it challenging to follow some of our hypotheses and suggestions. In response to this concern, we have prepared a detailed answer addressing each point raised by the reviewer. By providing thorough explanations and clarifications, we aim to improve the understanding of our paper and address any potential confusion. Once again, we sincerely appreciate the reviewer's efforts and feedback, which have undoubtedly contributed to the enhancement of our work.

Additional remarks

l.92 replace PSI by PSII - Corrected

L.161 replace In C. reinhardtii, PSII … by In C. reinhardtii, PsbP…. - Corrected

L.282 reveilles? reveals? - Corrected

L.162,163 “One example is a R368F substitution 16Å away of 167 the oxygen evolving complex (OEC). A nearby substitution of K371G might also contribute to the stability of this important site.” These substitutions refer presumably to C. reinhardtii vs C. ohadii. This should be stated more clearly in the text.

Answer:

Thank for the very careful reading, it was a mistake. We deleted these two sentences.

Fig. 3 Colors of the subunits and cofactors change from one panel to the other even in the same Figure. It would be less confusing for the reader to use the same color code for all Figures.

Answer:

We totally agree with this valuable comment, and new pictures with the same colours for the same subunits were done.

Comment:

Some abbreviations in the Figures need to be explained in the legends, e.g. PHO, BCT, PL, LHG ±

Answer:

The explanations are added to the pictures.

Reviewer 3 Report

The manuscript of Fadeeva at el. presents the CryoEM structure of the Photosystem II complex of the green alga Chlorella ohadii at 2.72 A resolution. Since the discovery of Chlorella ohadii understanding the mechanistic background of the very high light tolerance of this desert species poses a significant scientific challenge, which was approached here by the high resolution structure determination of the PSII complex. The structure represents the state of the art regarding the technical approach and reveals some interesting differences in subunit composition in comparison with other known PSII structures, especially of Chlamydomonas reinhardtii. The structural changes are located mostly at the donor side of PSII affecting the vicinity of the water-oxidizing complex, as well as in the acceptor side at the region of the QB binding site. Although the authors suggest to relate these changes to the high photo-tolerance of PSII the listed arguments are not convincing. Therefore, requires revision in the light of the following comments.

Comments:

1, The work presents a well resolved structure of PSII, which is partly different from structures obtained from other organisms, which are more vulnerable for light damage than Chlorella ohadii. This provides important structural information that could be used for identifying the structural basis, if any, of light tolerance. However, as described below the current manuscript could not present convincing ideas for structure-based understanding of photo-tolerance.

2, The argument that QB was not present in the previous structures is not correct since QB was retained in the PSII structure from Thermosynechococcus vulcanus (Umena et al. 2011, Nature) and also in other PSII from cyanobacteria.

3, The manuscript argues that the modified structure around the QB site facilitates fast electron transfer from QA- to QB, and from QB- towards the PQ pool which effects provide photoprotection via fast elimination of reduced quionone radicals in the PSII RC. Unfortunately, there are no experimental data in the literature that would support accelerated QA- to QB electron transfer rate in Chlorella ohadii. Also, the fast elimination of semireduced QB does not have experimental support. In all other organisms semireduced QB is strongly bound to the QB site and can stay there for a long time (tens of minutues, or longer), this is an essential part of the so-called two-electron gate mechanism at the PSII acceptor side. It would be very surprising if Chlorella ohadii would be different in that respect. The observation of the B thermoluminescence band from the S2QB- recombination (New Phytologist (2016) 210: 1229–1243) shows that QB- is stable under normal conditions in Chlorella ohadii, which contradicts the idea that the electron would be transferred rapidly from QB- to the PQ pool. This part of the text (lines 271-274) should be corrected.

4, The idea that the additional PQ molecules found close to QB could be involved in photoprotection is weakened by the presence of the very similar quinone binding site (Qc) in the cyanobacterial PSII structure which contains an additional quinone besides QB (Nat. Struct. Mol. Biol., 2009, 16, 334-342) although this cyanobacterium (Thermosynechococcus elongatus) is not phototolerant.

 5, The argument that the PQ molecule which is seen in the QB pocket is a tightly bound oxidized QB, is misleading. In thylakoids and intact algal cells ca. half of PSII contains a semireduced QB (QB-), which strongly binds to the QB site (in the other half of the centers the QB site is empty since oxidized PQ can easily leave the site). By CryoEM, or X-ray crystallography, it is impossible to say if QB is in the oxidized or semireduced form in the QB site, therefore it is most likely that the QB seen in the structure is actually QB- whose stability is a simple consequence of its semireduced state, and has nothing to do with phototolerance. The presence of stable QB- after single flash illumination was demonstrated by thermoluminescence measurements (the so called B band arises from the recombination of the S2 state with QB-) showing that QB- is similarly stable in Chlorella ohadii as in other organisms (New Phytologist (2016) 210: 1229–1243).

6, Most of the published data, including a recent paper (Photosynt. Res. (2021) 147:329–344)) assigns the photo-tolerance of Chlorella ohadii PSII to a cyclic electron flow inside PSII, which could harmlessly dissipate the longer-lived light-induced charge separated states before they could be involved in the formation of damaging radicals and/or reactive oxygen species. This PSII-CEF pathway was suggested to shortcut QB- and P680+ in Chlorella ohadii (BBA - Bioenergetics 1858 (2017) 873–883). It is not impossible that under high light conditions PSII-CEF is somehow induced, however, the presented structural data do not give useful information about the possible mechanism of this effect. If so, it should be commented in the text.

Author Response

Comments:

The manuscript of Fadeeva at el. presents the CryoEM structure of the Photosystem II complex of the green alga Chlorella ohadii at 2.72 A resolution. Since the discovery of Chlorella ohadii understanding the mechanistic background of the very high light tolerance of this desert species poses a significant scientific challenge, which was approached here by the high-resolution structure determination of the PSII complex. The structure represents the state of the art regarding the technical approach and reveals some interesting differences in subunit composition in comparison with other known PSII structures, especially of Chlamydomonas reinhardtii. The structural changes are located mostly at the donor side of PSII affecting the vicinity of the water-oxidizing complex, as well as in the acceptor side at the region of the QB binding site. Although the authors suggest to relate these changes to the high photo-tolerance of PSII the listed arguments are not convincing. Therefore, requires revision in the light of the following comments.

Answer:

We express our gratitude to the reviewer for their prompt and meticulous revision of our paper. We acknowledge that our intention to present the ideas in a concise form may have made it challenging to follow some of our hypotheses and suggestions. In response to this concern, we have prepared a detailed answer addressing each point raised by the reviewer. By providing thorough explanations and clarifications, we aim to improve the understanding of our paper and address any potential confusion.

Comment:

The work presents a well resolved structure of PSII, which is partly different from structures obtained from other organisms, which are more vulnerable for light damage than Chlorella ohadii. This provides important structural information that could be used for identifying the structural basis, if any, of light tolerance. However, as described below the current manuscript could not present convincing ideas for structure-based understanding of photo-tolerance.

Answer:

We appreciate the opportunity to discuss the summarized evidence in your paper, which aims to explain the exceptional photo tolerance of C. ohadii based on structural data. Let's address each point individually:

  1. The presence of additional tightly bound shielding subunits in the sensitive oxygen-evolving complex (OEX) of C. ohadii, which remain stable even during purification, sets it apart from similar PSII structures found in other organisms such as Chlamydomonas. This unique characteristic suggests that C. ohadii has a lower probability of sustaining damage even under strong light conditions, thereby increasing its photo tolerance.
  2. The additional protection provided to the cytochrome b559 by Psb10 and PsbY in C. ohadii plays a crucial role in minimizing the formation of harmful radicals that can rapidly damage PSII. This protective mechanism further contributes to the photo tolerance of C. ohadii.
  3. The unique QB-binding cavity found in C. ohadii is an intriguing feature that may also contribute to its exceptional photo tolerance. While the specific details of this explanation are likely discussed in your paper, it suggests that the distinct characteristics of the QB-binding cavity in C. ohadii enable more efficient and stable binding of the QB electron acceptor, which in turn influences the overall stability and functionality of PSII.

By summarizing these pieces of evidence, our paper aims to provide a comprehensive understanding of the structural factors that contribute to the exceptional photo tolerance observed in C. ohadii. We commend your efforts in analyzing and presenting these unique characteristics and their potential implications.

Comment:

The argument that QB was not present in the previous structures is not correct since QB was retained in the PSII structure from Thermosynechococcus vulcanus (Umena et al. 2011, Nature) and also in other PSII from cyanobacteria.

Answer:

We appreciate the reviewer's valuable notice. Based on the suggestion, we have made the necessary addition and clarification in the paragraph. The revised version now reads as follows:

"However, in contrast to most preparations of eukaryotic PSII, where QB is missing, in C. ohadii QB, it was clearly identified at its binding site. So far, QB was clearly seen mostly in significantly more stable PSII preparations from cyanobacteria" (lines 269-271).

With this addition, we aim to highlight the unique observation of QB being present at its binding site in C. ohadii PSII, which is not commonly observed in other eukaryotic PSII preparations. Additionally, the comparison to more stable PSII preparations from cyanobacteria provides further context to support the significance of this finding.

Comment:

The manuscript argues that the modified structure around the QB site facilitates fast electron transfer from QA- to QB, and from QB- towards the PQ pool which effects provide photoprotection via fast elimination of reduced quinone radicals in the PSII RC. Unfortunately, there are no experimental data in the literature that would support accelerated QA- to QB electron transfer rate in Chlorella ohadii.

Answer:

Thank you for giving us the opportunity to discuss our novel hypothesis in greater detail. We were mindful of achieving a balance between compactness and clarity in presenting our ideas. It's important to note that our hypothesis does not claim an increase in the maximal transfer rate from QA- to QB. Instead, it proposes a structure-based mechanism in which the reduced QB efficiently transfers electrons to secondary quinones that are in close proximity within the hydrophobic cavity. To provide further clarity, we have added a few lines to the manuscript (lines 282-286) explaining the feasibility and potential consequences of this mechanism. Only time and further research will reveal if our hypothesis is accurate. 

Comment:

 There are no experimental data in the literature

Answer:

we quote papers [36] and [37] supporting possibility of such mechanism. We are sure that our group or other researchers in the next few years will find a convincing way to demonstrate this experimentally.

 Comment:

Also, the fast elimination of semireduced QB does not have experimental support. In all other organisms semireduced QB is strongly bound to the QB site and can stay there for a long time (tens of minutes, or longer), this is an essential part of the so-called two-electron gate mechanism at the PSII acceptor side. It would be very surprising if Chlorella ohadii would be different in that respect. The observation of the B thermoluminescence band from the S2QB- recombination (New Phytologist (2016) 210: 1229–1243).  shows that QB- is stable under normal conditions in Chlorella ohadii, which contradicts the idea that the electron would be transferred rapidly from QB- to the PQ pool. This part of the text (lines 271-274) should be corrected.

Answer:

We think, that mentioned experiments could not differentiate between QB and closely situated quinones. Thus, they can not be used to rule out our hypothesis. We are actively looking for conditions that will keep those quinones in the pocket.     

Comment:

The idea that the additional PQ molecules found close to QB could be involved in photoprotection is weakened by the presence of the very similar quinone binding site (Qc) in the cyanobacterial PSII structure which contains an additional quinone besides QB (Nat. Struct. Mol. Biol., 2009, 16, 334-342) although this cyanobacterium (Thermosynechococcus elongatus) is not phototolerant.

Answer:

The presence of additional quinone (Qc) reported in this manuscript was structurally challenged (please, look at the reported electron density). 

 Comment:

The argument that the PQ molecule which is seen in the QB pocket is a tightly bound oxidized QB, is misleading. In thylakoids and intact algal cells ca. half of PSII contains a semireduced QB (QB-), which strongly binds to the QB site (in the other half of the centers the QB site is empty since oxidized PQ can easily leave the site). By CryoEM, or X-ray crystallography, it is impossible to say if QB is in the oxidized or semireduced form in the QB site, therefore it is most likely that the QB seen in the structure is actually QB- whose stability is a simple consequence of its semireduced state, and has nothing to do with phototolerance. The presence of stable QB- after single flash illumination was demonstrated by thermoluminescence measurements (the so-called B band arises from the recombination of the S2 state with QB-) showing that QB- is similarly stable in Chlorella ohadii as in other organisms (New Phytologist (2016) 210: 1229–1243).

Answer:

We think, that mentioned experiments could not differentiate between QB and closely situated quinones. Thus, they cannot be used to rule out our hypothesis. We are actively looking for conditions that will keep those quinones in the pocket.    

Comment:

Most of the published data, including a recent paper (Photosynt. Res. (2021) 147:329–344)) assigns the photo-tolerance of Chlorella ohadii PSII to a cyclic electron flow inside PSII, which could harmlessly dissipate the longer-lived light-induced charge separated states before they could be involved in the formation of damaging radicals and/or reactive oxygen species. This PSII-CEF pathway was suggested to shortcut QB- and P680+ in Chlorella ohadii (BBA - Bioenergetics 1858 (2017) 873–883). It is not impossible that under high light conditions PSII-CEF is somehow induced, however, the presented structural data do not give useful information about the possible mechanism of this effect. If so, it should be commented in the text.

Answer:

The structure was solved only for low-light grown algae cells and still provides structural details mentioned in response to point #1.  We also added more details to our explanations (lines 234-242). “PsbY joined a triplet of transmembrane helices that were formed by PsbE, PsbF, and Psb10, while running antiparallel to them (Figure 3c). Together, they shield the heme group of cytochrome b559, which otherwise would be exposed to the membrane interior environment and could easily contact with potential deleterious electron acceptors. Thus, PsbY and Psb10 provide additional protection to that of PsbE and PsbF. This arrangement stabilized the position of the prosthetic group that might be crucial for the protection of PSII from photodamage caused by excessive light intensities, when electron acceptors are lacking. So, even increased CEF during high light illumination does not lead to increased production of free radicals and excess damage of PSII with subsequent inactivation”.

Round 2

Reviewer 2 Report

show a supplementary Fig showing side by side the structures of the two algae and pinpointing the differences.

The revised version has been improved although as I suggested earlier a supplementary Fig showing side by side the  PSII structures of C. ohadii and C reinhardtii and pinpointing the differences would have been helpful.

The English is still bumpy in some parts. Ideally the text should be revised by a person whose mother tongue is English.

Minor comments   

L. 385     delete “was”

L.730  * leaving organisms” what does this mean? Green organisms?

Author Response

Thank you for your remarks.

Comment:

The revised version has been improved although as I suggested earlier a supplementary Fig showing side by side the PSII structures of C. ohadii and C reinhardtii and pinpointing the differences would have been helpful. –

Answer

Unfortunately, we are not sure such representation can be made clear and easy to follow. Both supercomplexes are huge and share too many common subunits. We changed Figure A3 and legend (marked in red) and offer to add next summarizing paragraph to the main text where it is now (lines 122-129). The added subunits are present on the background of Chlamydomonas PSII (PDB 6KAD) on which the Ohadii PSII was superimposed.

The English is still bumpy in some parts. Ideally the text should be revised by a person whose mother tongue is English.

Minor comments   

  1. 385     delete “was” – Done

L.730  * leaving organisms” what does this mean? Green organisms? – corrected to “living organism”

Reviewer 3 Report

The manuscript was improved as a result of the revision, and as a whole it represents important data about the PSII structure in Chlorella ohadii.

It still has problems with assigning the extreme photoprotection to fast electron transfer from QB- to the PQ pool, which contradicts the accepted two-electron gate mechanism of electron transfer at the PSII acceptor side, whose essential part is the presence of semireduced QB in the QB site until the next electron arrives from QA-. The argument of the authors, in their reply to my first review, claiming that the electron arriving from QA- may be stabilized on a secondary quinone in the cavity instead of QB-, could be correct. However, this electron is able to recombine with the S2(S3) state of the water oxidizing complex according to earlier thermoluminescence data, so it is unclear why would it be less damaging for PSII activity if the electron is sitting on another PQ in the QB pocket instead of QB-.

It is not impossible, although highly unlikely, that the two-electron gate  mechanism is not valid for Chlorella ohadii. However, this idea represents a testable hypothesis which could be clarified in later studies.

Author Response

Thank you for your remarks.

The manuscript was improved as a result of the revision, and as a whole it represents important data about the PSII structure in Chlorella ohadii.

It still has problems with assigning the extreme photoprotection to fast electron transfer from QB- to the PQ pool, which contradicts the accepted two-electron gate mechanism of electron transfer at the PSII acceptor side, whose essential part is the presence of semireduced QB in the QB site until the next electron arrives from QA-. The argument of the authors, in their reply to my first review, claiming that the electron arriving from QA- may be stabilized on a secondary quinone in the cavity instead of QB-, could be correct.

However, this electron is able to recombine with the S2(S3) state of the water oxidizing complex according to earlier thermoluminescence data, so it is unclear why would it be less damaging for PSII activity if the electron is sitting on another PQ in the QB pocket instead of QB-.

Answer:

We think that it decreases the probability of recombination.

It is not impossible, although highly unlikely, that the two-electron gate mechanism is not valid for Chlorella ohadii. However, this idea represents a testable hypothesis which could be clarified in later studies.

Answer:

We are grateful for the comments and discussion which will be very useful during planning of experiments.